# Depression and Sleep

**DOI:** 10.3390/ijms20030607

**Published:** 2019-01-31

**Authors:** Axel Steiger, Marcel Pawlowski

**Affiliations:** 1Max Planck Institute of Psychiatry, Research Group Sleep Endocrinology, 80804 Munich, Germany; p@wlowski.de; 2Centre of Mental Health, 85049 Ingolstadt, Germany

**Keywords:** depression, sleep, sleep EEG, biomarkers, antidepressants, cordance

## Abstract

Impaired sleep is both a risk factor and a symptom of depression. Objective sleep is assessed using the sleep electroencephalogram (EEG). Characteristic sleep-EEG changes in patients with depression include disinhibition of rapid eye movement (REM) sleep, changes of sleep continuity, and impaired non-REM sleep. Most antidepressants suppress REM sleep both in healthy volunteers and depressed patients. Various sleep-EEG variables may be suitable as biomarkers for diagnosis, prognosis, and prediction of therapy response in depression. In family studies of depression, enhanced REM density, a measure for frequency of rapid eye movements, is characteristic for an endophenotype. Cordance is an EEG measure distinctly correlated with regional brain perfusion. Prefrontal theta cordance, derived from REM sleep, appears to be a biomarker of antidepressant treatment response. Some predictive sleep-EEG markers of depression appear to be related to hypothalamo-pituitary-adrenocortical system activity.

## 1. Introduction

Insomnia is a frequent symptom of depression. Conversely, it is a risk factor for the development of a depressive episode [1]. Objective sleep is assessed by polysomnography, also named sleep electroencephalogram (EEG). Sleep EEG appears to be suitable method of gaining biomarkers of depression, and these biomarkers may contribute to the nosology, prognosis, and prediction of therapy response in depression.

There are two reasons why psychiatrists became interested in sleep EEG in the 1970s. Rapid eye movement (REM) latency was suggested to indicate depression [2]. Furthermore, it was found that most antidepressants suppressed REM sleep [3]. Previously it was thought that REM latency may distinguish between certain subtypes of depression. In addition, it was hypothesized that REM suppression is the mechanism of action of antidepressants [4]. Since then, research has provided much more complex results.

Mammalian sleep consists of alternating periods of REM and non-REM sleep. Infants sleep in a polyphasic fashion. During human development, a mostly monophasic sleep-wake pattern emerges. The criteria by Rechtschaffen and Kales [5] differentiate between four stages of non-REM sleep, whereas the more recent classification by the American Academy of Sleep Medicine (AASM) shows three stages [6]. Stages 3 and 4 according to Rechtschaffen and Kales, or N3 according to AASM, are also termed slow-wave sleep (SWS). Shortly after going to bed, young normal subjects enter the lighter sleep stages N1 and N2 of non-REM sleep, followed by N3 (SWS). The major portion of SWS occurs during the first non-REM period. After a mean duration of 90 min of the first non-REM period, the first REM period occurs. The first REM period is relatively short; but its duration consecutively increases during the night. Accordingly, during the first half of the night, SWS preponderates. During the second half of the night, however, stage N2 and REM sleep dominate. Most subjects show four to five sleep cycles during the night, each consisting of one period of non-REM sleep and one period of REM sleep.

## 2. Sleep EEG in Patients with Depression

Most patients with depression suffer from impaired sleep, about 80 percent suffer from insomnia and 15–35 percent from hypersomnia [7,8]. Patients with depression show characteristic sleep-EEG changes [8,9,10] including:(i)Impaired sleep continuity (prolonged sleep latency, increased intermittent awakenings, early morning awakenings).(ii)Disinhibition of REM sleep: shortened REM latency, or sleep onset REM periods (SOREMs, REM latency 0–20 min), prolonged first REM period, enhanced REM density (measure of frequency of rapid eye movements) particularly during first REM period.(iii)Changes in non-REM sleep (decreased stage N2 and SWS, in younger patients shift of SWS from the first to the second sleep cycle).

Patients with depression show reduced EEG delta power, also termed slow wave activity (SWA) throughout the night [11,12,13,14]. Sleep EEG is modulated by age and gender in healthy volunteers and in depressed patients as well. In the third decade of the life, SWS and SWA start to decrease. Menopause is the major turning point in sleep quality in women. In male subjects, however, sleep quality declines continuously during aging. In patients with depression, age and illness exert a synergistic effect on sleep EEG. The effect of aging on sleep EEG in patients with depression and normal control subjects were investigated in two studies [15,16]. These studies showed a clear effect of age on REM latency, whereas patients and healthy subjects did not differ until the middle of the fourth decade. On the other hand, REM density was enhanced in all investigated age groups in patients when compared with controls. SWS declined throughout the lifespan without differences between patients and controls.

In two longitudinal studies, no changes in sleep-EEG variables of depressed patients were found between acute depression and remission [17,18]. In one study, sleep stage 4 decreased after remission when compared to baseline [18]. Similar results were reported in depressed adolescents [19]. Increasing abnormality of REM sleep variables was observed during a long-term study on repetitive episodes of depression. SWS did not differ between episodes [20].

The view that shortened REM latency is a specific marker of depression [2] was challenged by other studies reporting similar changes in other psychiatric disorders including mania [21], schizophrenia [22], schizoaffective disorder [23], obsessive-compulsive disorder [24], panic disorder [25], eating disorders [26], and sexual impotence [27]. The finding that sleep-EEG changes persist in remitted patients [17,18] may explain that comorbidity with depression or a history of depression result in a shortened REM latency in these disorders. This view is supported by two studies by Lauer et al. [28,29]. These authors compared three groups of patients with major depression, anorexia nervosa, and bulimia with healthy subjects. The latter two groups of patients were never depressed. REM density was enhanced in the patients with depression [28]. In the other study, depressed patients, patients with panic disorder without a history of depression, and normal controls were compared. Differences were observed during the first sleep cycle. SWS was reduced and REM time and REM density were increased during this interval in the depressed patients. In patients with panic disorder this cycle was shortened. REM latency was shorter in both groups of patients than in healthy controls [29].

## 3. Sleep EEG in High-Risk Probands for Affective Disorders

In the Munich Vulnerability Study on affective disorders, a prospective high-risk design was applied. In order to identify premorbid vulnerability factors for affective disorder, high-risk probands were examined. They had a high genetic load for affective disorders due to a positive family history. Comparison of the high-risk probands with healthy subjects without a family history for this disease showed enhanced REM density and reduced time spent in non-REM sleep during the first sleep cycle [30]. This finding remained stable at the follow-up investigation four years later [31]. In a subgroup of the high-risk probands, the cholinergic REM sleep induction test was performed using the cholinomimetic RS86. At baseline, REM latency did not differ between high-risk probands and the controls. After RS86, REM latency was decreased in the high-risk probands [32]. This finding points to a threshold cholinergic dysfunction in the high-risk probands. The response pattern in the cholinergic REM sleep induction test predicted the onset of the first episode of depression [33]. Twenty subjects of the initial sample of 83 high-risk probands of this study developed an affective disorder during the follow-up period. In these subjects, the premorbid sleep EEG showed increased REM density during the total night and during the first REM period when compared to healthy volunteers [34]. These findings show that increased REM density meets all requirements for biological vulnerability markers of affective disorders. The authors recommend REM density as a possible endophenotype in family studies [34].

## 4. Sleep EEG and Risk Genes for Depression

*P2RX7* is a susceptibility gene for affective disorders. It is located on chromosome 12 q24, which appears to be associated with major depression [35] and bipolar disorder [36]. *P2RX7* is found in immune, endothelial, and epithelial cells, and regulates various aspects of immune function, as expression and secretion of cytokines [37]. The single nucleotide polymorphism (SNP) rs2230912 in the *P2RX7* gene (base change 1405A>G) leading to substitution of glutamine (Gln, Q) by arginine (Arg, R), at codon 460 (Gln 460 Arg, Q 460 R), has been associated with mood disorders [38,39,40]. To clarify whether elevated risk for depression related to this SNP shows sleep-EEG changes, young healthy volunteers who were free of psychiatric disorders in their own and family history, were investigated in the sleep laboratory. Homozygous (A/A) subjects and heterozygous (A/G) carriers of the risk variant were compared. Significant differences in sleep-EEG were found between groups. In the heterozygous (A/G) subjects, prolonged sleep latency and shortened sleep period time was found; the number of entries from stage N2 into N1 and wakefulness was enhanced during the first sleep cycle; in the lower spindle range frequencies were elevated, particularly in parietal regions; peak frequencies of all sleep spindles were lower during non-REM sleep. In particular, elevated parietal variations during stage N2 beta frequencies were reported. These data show that healthy volunteers with a potential risk for affective disorders related to their *P2RX7* genotype differ in sleep EEG from subjects with lower risk [41].

Mice that harbor *P2RX7*-Gln 460 AG and the wild-type *P2RX7* showed, compared to homozygous *P2RX*7 wildtype and *P2RX7*^hQ460R^ mice an increase of entries to REM sleep during the light period, suggesting a stronger drive towards REM sleep and more fragmented sleep cycles. Furthermore, SWA was lower and the amount of deep non-REM sleep was only small in heterozygous mice. Taken together, heterogeneous mice show altered sleep architecture and reduced sleep quality compared to homozygous mice [41].

## 5. Effects of Antidepressants on Sleep EEG

Most antidepressants suppress REM sleep in patients and in healthy volunteers. REM suppression includes prolonged REM latency, reduced time spent in REM sleep, and decreased REM density. Withdrawal of REM suppressing antidepressants is followed by REM rebound. Decreased REM latency, increased REM time, and enhanced REM density are the components of REM rebound. All these variables exceed baseline values. Withdrawal of antidepressants after two weeks of treatment prompted a REM rebound that persisted after one week [42]. REM suppression occurs after tricyclics [43,44], tetracyclics [3], selective serotonin reuptake inhibitors (SSRIs) [45,46], selective noradrenaline reuptake inhibitors (NRI) [47], selective serotonin and noradrenaline reuptake inhibitors (SNRI) [48], reversible [49,50,51], and short acting reversible [52] monoamine oxidase inhibitors. Only some antidepressants do not suppress REM sleep including trimipramine [53], bupropion [54], the serotonin reuptake enhancer tianeptine [55], and the noradrenergic and specific serotonergic antidepressant (NaSSA) mirtazapine [56,57].

Various antidepressants differ in the potency to suppress REM sleep. Total REM suppression was found after clomipramine [58] and the irreversible monoamine oxidase inhibitors phenelzine and tranylcypromine [59]. Additionally, distinct REM suppression was observed in healthy volunteers following the combined SSRI and serotonin 5-HT1A receptor agonist vilazodone [60]. Also the dosage and plasma concentrations of the substances influence the amount of changes in REM sleep [58].

After selective REM sleep deprivation, but not after selective non-REM sleep deprivation for three weeks, antidepressant effects were observed [4]. This finding and the observation that most antidepressants suppress REM sleep resulted in the hypothesis that REM suppression is the mechanism of action of antidepressant drugs. This theory however was challenged by the lack of antidepressant effect of selective REM suppression for the first eleven days of treatment [61]. In addition, the fact that some antidepressants do not suppress REM sleep, like trimipramine, tianeptine, and mirtazapine, contradicts the hypothesis by Vogel et al. [4].

The comparison of the effects of the stereoisomeres of oxaprotiline, R(−)oxaprotiline, and S(+)oxaprotiline on sleep support the view that REM suppression is a distinct, but not absolute requirement for antidepressant effects of a substance. S(+)oxaprotiline suppressed REM sleep in patients with depression, whereas R(−)oxaprotiline did not share this effect. S(+)oxaprotiline had better antidepressant effects then R(−)oxaprotiline [62]. The effects of most antidepressants on REM sleep are similar. In contrast, substances differ in their effect on sleep continuity and on non-REM sleep. Whereas most tricyclics elevate SWS [3], clomipramine [58], and imipramine [53] diminished SWS. SSRIs do not modulate SWS, but impair sleep continuity and enhance intermittent wakefulness [63,64]. In addition, the NaRI reboxetine diminishes sleep efficiency and elevates intermittent wakefulness and stage 2 sleep [47]. After vilazodone, REM sleep in healthy volunteers was distinctly suppressed together with increases in SWS and SWA in the first and the last third of the night [60]. After the SSNRI duloxetine, stage 3 increased in depressed patients [48]. On the second day of mirtazapine treatment, patients with depression showed an increase in total sleep time and sleep efficiency and a decrease in time awake. These effects persisted after four weeks, when SWS, low delta, theta, and alpha activity increased [57]. After two days of treatment with amitriptyline, the increase seen in REM latency correlated with the clinical outcome after four weeks [65]. A single observation was reported for imipramine [53], but not after clomipramine [66].

## 6. Contribution of the HPA System to Sleep-EEG Abnormalities in Depression

It is well established that over-activity of the hypothalamo-pituitary-adrenocortical (HPA) system plays a key role in the pathophysiology of affective disorders [67]. In two longitudinal studies, nocturnal cortisol [18,68] and ACTH [68] concentrations were compared between acute depression and recovery. In comparison to healthy controls, cortisol and ACTH levels were elevated in patients with acute depression [68]. After treatment with electroconvulsive therapy or amitriptyline and remission, ACTH levels decreased [68]. Similarly, comparison of cortisol levels between acute depression and recovery in patients who were drug-free at both examinations, showed a decrease in cortisol levels [18]. These findings show that enhanced nocturnal HPA hormone secretion is a state marker of acute depression. Administration of the key hormone of the HPA system, corticotropin-releasing hormone (CRH), prompted more shallow sleep in rats [69], rabbits [70], and mice [71,72]. Similarly, after repetitive intravenous (iv) injections of CRH around sleep onset, SWS decreased and endocrine changes that are characteristic for depression (i.e., elevated cortisol levels, blunted growth hormone (GH) peak) were observed in young male volunteers [73]. Mouse mutants overexpressing CRH in the entire central nervous system or only in the forebrain showed increased REM sleep compared to wild-type mice [74].

In healthy women, the effects of pulsatile CRH injections on sleep EEG were more distinct than in healthy males, as intermittent wakefulness increased during the total night and the sleep efficiency index decreased. Furthermore, during the first third of the night, REM sleep and stage 2 sleep increased and sleep stage 4 was diminished. Cortisol levels were elevated throughout the night, whereas GH secretion remained unchanged [75].

Already in kindergarten children, associations were found between unfavorable sleep-EEG patterns, elevated HPA activity, and more difficult behavioral psychosocial dimensions [76]. In preschool children, sleep EEG was recorded and saliva samples were collected after awakening and before and after a psychological challenge for cortisol analysis. Children labeled as “poor” sleepers showed significantly increased morning cortisol values in comparison to “good” sleepers. Increased cortisol values after stress were significantly associated with an increased number of awakenings after sleep onset and an increased amount of sleep stages 1 and 2. Furthermore, psychological difficulties, such as impulsivity, over-anxiousness, and social inhibition, showed a significant association with low sleep efficiency.

In a clinical trial of the CRH receptor type 1 (CRHR1) antagonist R121919, a random subgroup of 10 patients had their sleep EEG assessed. Sleep-EEG recordings were performed at baseline, before treatment, after one week of active treatment, and at the end of the fourth week of treatment. SWS increased after week 1 and after week 4 compared to baseline. During the same period, the number of awakenings and REM density decreased. Separate evaluation of these changes for two different dose ranges showed no significant effects with the lower dose, whereas with the higher dose, REM density decreased, and SWS increased significantly between baseline and week 4. Positive associations were found between the Hamilton-Depression-Score and SWS at the end of active treatment. These results support the hypothesis that CRH is involved in the pathophysiology of sleep-EEG changes in depressed patients. In addition, these findings suggest that CRHR1 antagonism induces normalization of the sleep EEG in depressed patients [77].

Multiple sclerosis patients receiving subchronic administration of the synthetic glucocorticoid receptor agonist methylprednisolone showed similar sleep-EEG changes as in patients with depression. These changes included shortened REM latency, enhanced REM density, and shift of SWS and SWA from the first to the second non-REM period [78].

In male human subjects and in male animals, GH-releasing hormone (GHRH) exerts effects on sleep that are opposite to those of CRH. SWS increases after intracerebroventricular (icv) administration of GHRH in rats [69,79], after injection into the medial preoptic area of rats [80], and after iv administration to rats [81]. Similarly in young male healthy volunteers, in a protocol analogous to the study by Holsboer et al. [73] repetitive iv administration of GHRH increased SWS and GH and decreased cortisol [82]. In women, however, sleep was impaired after GHRH and cortisol and ACTH was enhanced [83,84], which is similar to the effects of CRH [73]. It is thought that at least in male patients a balance exists between GHRH and CRH in sleep regulation. GHRH appears to be active at the beginning of the night as mirrored by the high amounts of SWS and GH. During the second half of the night, CRH appears to preponderate and to induce more REM sleep and elevated cortisol. During depressive episodes (and during normal aging as well) the GHRH/CRH ratio is changed in favor of CRH due to CRH overactivity in affective disorders (or to declining GHRH activity during aging) (see Figure 1). A synergism of CRH and cortisol may contribute to REM sleep disinhibition.

## 7. Amyloid-β and Sleep

Aggregation and accumulation of amyloid-β (Aβ) contributes to the development of Alzheimer’s disease [85]. Several recent studies address the interaction of Aβ and sleep. Using positron emission tomography Shokri-Kojori et al. (2018) showed significant increases in Aβ burden in the right hippocampus and thalamus after a night of sleep deprivation in healthy controls. These increases were associated with worsening of mood after sleep deprivation [86]. In rats, sleep deprivation impaired cognitive function and elevated Aβ levels [87]. The effect of sleep on overnight cerebrospinal fluid (CSF) Aβ kinetics was tested in healthy volunteers using intracerebroventricular (icv) lumbar catheters for serial sampling of CSF while subjects were sleep deprived, received sleep promoting sodium oxybate or slept normally. To measure Aβ kinetics all participants were infused with ^13^C_6_-leucine. Sleep deprivation increased overnight Aβ38, Aβ40, and Aβ42 levels by 25–30% via increased overnight Aβ production relative to sleeping subjects. The authors concluded that disrupted sleep increases Alzheimer’s disease risk by increased Aβ production [88]. In order to elucidate whether chronic sleep restriction potentiates the brain impact of Aβ oligomers (AβOs) studies in mice were performed. A single icv infusion of Aβ oligomers disturbed sleep pattern in mice. Conversely, chronically sleep restricted mice showed higher brain expression of pro-inflammatory mediators, reduced levels of pre- and post-synaptic marker proteins. Furthermore, this study exhibited increased susceptibility to the sub-toxic dose of AβOs on performance in a novel object recognition memory task. After sleep restriction, elevated brain tumor necrosis factor α (TNF-α) levels were found in response to AβOs. Neuronal impairment in sleep restricted AβOs infused mice was prevented by a TNF-α neutralizing monoclonal antibody. The authors discuss a dual relationship between sleep and Alzheimer’s disease with disruption of sleep wake patterns by AβOs and increased brain vulnerability to AβOs after chronic sleep restriction [89]. In Alzheimer’s disease model mice, chronic sleep fragmentation was induced by a running-wheel-based device that resulted in increased Aβ deposition in the mouse brain. The severity of Aβ deposition showed a significant positive correlation with the extent of sleep fragmentation [90]. Specific disruption of SWA in healthy adults without sleep disorders correlated with an increase in Aβ [91]. In patients with insomnia CSF Aβ levels were significantly higher than in healthy controls [92].

Interestingly, there is some overlap between the pathophysiology of depression, Alzheimer’s disease and sleep. Human neuroblastoma cells produced more Aβ after treatment with CRH [93]. Morgese et al. (2017) discuss that chronic stress may represent common biological bases linking Alzheimer’s dementia and depression [94]. The interaction of sleep and Aβ in patients with depression is an open topic on the research agenda.

## 8. State and Vulnerability Markers Related to Antidepressant Therapy

In a clinical trial, the effects of the serotonin reuptake enhancer tianeptine and the SSRI paroxetine were compared. The effects of these substances on sleep EEG were investigated in a subgroup of these patients. Sleep EEG was recorded at days 7 and 42 after the start of treatment with either substance. In male treatment responders, a distinct decline in the higher sigma frequency range (14–16 Hz) during non-REM sleep was found independently of medication. In contrast, male and female non-responders did not show marked changes in this frequency range. This finding supports the view that gender should be taken into account when the biological effects of drugs are studied. After paroxetine, the amount of REM sleep was reduced and intermittent wakefulness was increased in comparison to tianeptine. In the total sample after one week of treatment, REM density was a predictor of treatment response. The change in REM density showed an inverse correlation to changes in the Hamilton Depression Score in the patients who received paroxetine, but not in those who received tianeptine [55].

Patients with depression who had participated in an earlier study with trimipramine were involved in an exploratory follow-up study. The retrospectively-assessed long-term course of depression in these patients was related with sleep-EEG variables during the acute episode. The lower the sleep continuity (total sleep time, sleep efficiency index, time spent awake, number of awakenings), the higher was the number of previous episodes of depression. This association disappeared at the end of drug treatment with a distinct association found between reduced SWS, particularly during the first third of the sleep period, elevated REM density (by trend), and the number of previous episodes. A clear association was observed between the prospective long-term course and sleep EEG, as increased REM density and decreased SWS at the end of treatment were associated with an elevated recurrence rate between the end of the trial and the follow-up study. These sleep-EEG variables showed an association with impaired HPA system, evident by abnormal results of the dexamethasone/CRH (DEX/CRH) test. Patients with an unfavorable long-term course of depression appear to show increasing aberrant sleep regulation. These changes seem predictive not only for treatment response during the acute episode, but also for recurrences in the long-term. These predictive sleep-EEG markers may relate with HPA system activity, since the more sleep-EEG markers were disturbed, the more the HPA system was impaired [95].

## 9. Cordance Derived from REM Sleep as a Predictor of Therapy Response

Cordance is a quantitative EEG measure that combines information from absolute and relative EEG spectral power. It correlates with regional brain activity. Theta frequency band of cordance shows positive correlation with cerebral blood perfusion [96]. Prefrontal theta cordance, derived from the awake EEG, correlates with antidepressant treatment outcome. After one week of drug treatment, prefrontal theta cordance decreased in several studies, irrespective of the investigated drugs [97,98,99,100]. It is thought that prefrontal theta cordance reflects activity of prefrontal cortex and anterior cingulate cortex (ACC) [101]. Both appear to be crucially involved in major depression [102]. During REM sleep, ACC activity is maximal. In contrast, the surrounding frontal cortex activity is minimal [103,104]. During REM sleep, ACC shows distinct oscillatory activity in the theta frequency band [105]. Therefore, prefrontal theta cordance is an ideal way to detect theta frequency band. Prefrontal theta cordance of depressed patients was measured during tonic REM sleep. In responders (of totally 20 depressive in-patients on various antidepressants), prefrontal theta cordance was significantly higher after the first week of antidepressant medication than in non-responders. This result was still significant after controlling for age, gender, and the number of previous episodes of depression. In addition, prefrontal cordance in all patients showed a significant positive correlation with the improvement of the Hamilton Depression Score between inclusion week and the first week of drug treatment [106].

## 10. Heart Rate Variability Derived from REM Sleep in Depressed Patients

The study by Adamczyk et al. [106] was extended to test whether heart rate variability (HRV) derived from REM sleep could represent a biomarker of antidepressant treatment response. A meta-analysis showed that major depression is associated with blunted HRV [107]. It was expected that an association of HRV and depression would be stronger in offline conditions like sleep. In patients with depression, HRV was derived from 3 min artefact free electrocardiogram sequence during REM sleep. In comparison to controls, HRV during REM sleep was decreased in depressed patients (responders as well as non-responders) during the fourth week of treatment in comparison to controls (see Figure 2). It showed a negative correlation with REM density in healthy subjects and in patients at week four. HRV derived from REM sleep appears to categorize healthy subjects and patients with depression [108].

## 11. Perspectives

This review presents sleep EEG as a promising tool for psychiatric research and clinical application in affective disorders.

The observation of subtle influence of the *P2RX7* genotype on sleep-EEG pattern should be extended to studies of the association of other risk genes of depression on sleep EEG in healthy and in depressed patients. This approach may support the efforts to establish a new nosology of depression related to neurobiology.

Cordance appears to help to differentiate early during treatment between responders and non-responders to antidepressant therapy. The next step will be to test the capacity of cordance to shorten the long way to recovery that many patients with depression suffer. This would be possible if the expected response to a certain antidepressant in a patient is tested using cordance after one week of treatment. If non-response is predicted, medication could be changed much earlier than in the traditional way of assessing response related to psychopathology after about four weeks.

Some antidepressants promote, and others impair sleep. However, it is not yet clear whether stability of remission is influenced by such differences in medication.

## Figures and Tables

**Figure 1 ijms-20-00607-f001:**
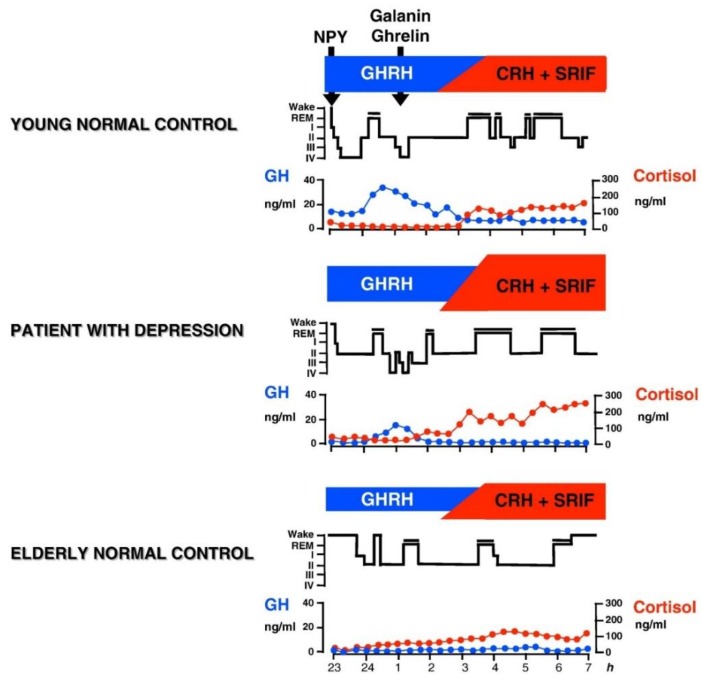
Patterns of normal and impaired peptidergic sleep regulation. Characteristic hypnograms and patterns of cortisol and GH secretion are shown in a young and in an elder healthy subject and in a patient with depression It is thought that GHRH is active during the first hours of sleep resulting in GH peak and the major portion of SWS during the night. During the second half of the night the influence of CRH preponderates which prompts increases of cortisol and REM sleep. Galanin and ghrelin may act as co-factors of GHRH. Somatostatin (SRIF) may impair sleep. The balance between GHRH and CRH changes during normal aging, when GHRH activity declines and during depressive episodes, when CRH activity is enhanced. Reprinted with permission from Springer, Nervenarzt, Schlafendokrinologie, Axel Steiger, 1995.

**Figure 2 ijms-20-00607-f002:**
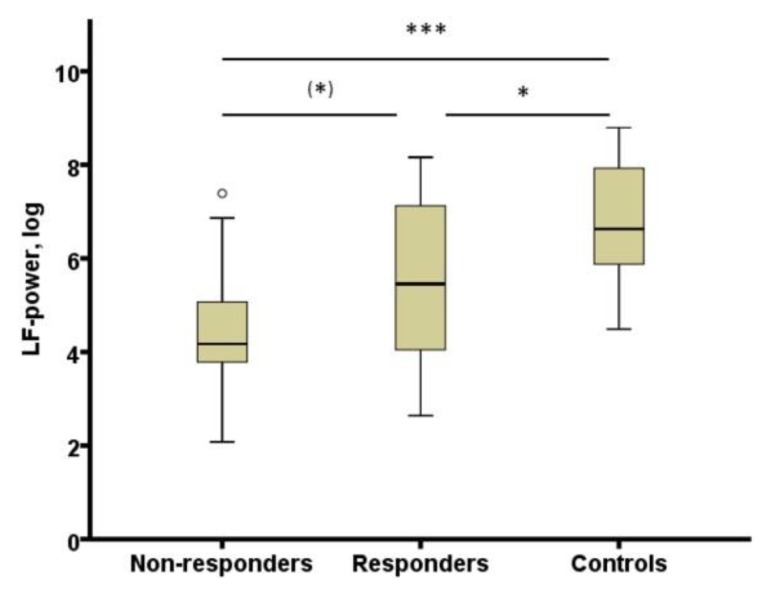
HRV (LF-power, log) in non-responders and responders at week 4 and controls. HRV: heart rate variability; LF-power, log: power in low frequency range (0.04–0.15 Hz) transformed with natural logarithm, (*) *p* < 0.10; * *p* < 0.05; *** *p* < 0.001. From [108] with permission from Elsevier.

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
