# Peer review of "Depression and Sleep"

_ijms, 2019, doi:10.3390/ijms20030607_

Round 1

Reviewer 1 Report

The review entitled "Depression and Sleep. " by Steiger and Pawlowski is that impaired sleep is a very important marker and also a factor for depression and the review gives the Readers of our Journal much information about it. The reviewer thinks it is suitable for publishing besides several points.

Points:

1)               There are many abbreviations in this manuscript, for example, SWS, SWA etc. These may be usual words for sleep studies, but some researchers may not be familiar with them. Please make abbreviation parts in the manuscript.  

2)               Are the abbreviations popular in selective noradrenaline reuptake inhibitors (NaRIs), selective serotonin and noradrenaline reuptake inhibitors (SSNRIs)? 

Author Response

We are glad about the favorable comment by this reviewer.

Reply to reviewer’s points:

1.)    There are many abbreviations in this manuscript, for example, SWS, SWA etc. These may be usual words for sleep studies, but some researchers may not be familiar with them. Please make abbreviation parts in the manuscript.  

Abbreviations are already explained in the text. The full word is given followed by the abbreviation in parentheses. Nevertheless, we have prepared a list of abbreviations. We ask the editor to decide whether this list should be published in addition.

List of abbreviations

Abbreviation

Full word

Definition

REM density

Rapid eye movement density

Measure of frequency of   rapid eye movements

REM latency

Rapid eye movement latency

Interval between sleep   onset and first epoch of sleep containing REM sleep

REM sleep

Rapid eye movement sleep

SWA

Slow wave activity

EEG delta power

SWS

Slow wave sleep

Also termed sleep stage N3

SOREM

Sleep onset rapid eye   movement period

REM period, 0 – 10 min   after sleep oset

2.)    Are the abbreviations popular in selective noradrenaline reuptake inhibitors (NaRIs), selective serotonin and noradrenaline reuptake inhibitors (SSNRIs)? 

In the English scientific literature mostly NRI is used instead of NaRI, and SNRI is often used instead of SSNRI Therefore, these abbreviations were changed in the manuscript.

Reviewer 2 Report

The review of Steiger and  Pawlowski is very interesting and of impact in the field.

However, I suggest to add another paragraph in regard to amiloyd levels after sleep deprivation (1: Rasmussen MK, Mestre H, Nedergaard M. The glymphatic pathway in neurological disorders. Lancet Neurol. 2018 Nov;17(11):1016-1024. doi: 10.1016/S1474-4422(18)30318-1. Review. PubMed PMID: 30353860; PubMed Central PMCID: PMC6261373.; 2: Brzecka A, Leszek J, Ashraf GM, Ejma M, Ávila-Rodriguez MF, Yarla NS, Tarasov VV, Chubarev VN, Samsonova AN, Barreto GE, Aliev G. Sleep Disorders Associated With Alzheimer's Disease: A Perspective. Front Neurosci. 2018 May 31;12:330. doi: 10.3389/fnins.2018.00330. eCollection 2018. Review. PubMed PMID: 29904334; PubMed Central PMCID: PMC5990625; 3: Shokri-Kojori E, Wang GJ, Wiers CE, Demiral SB, Guo M, Kim SW, Lindgren E, Ramirez V, Zehra A, Freeman C, Miller G, Manza P, Srivastava T, De Santi S, Tomasi D, Benveniste H, Volkow ND. β-Amyloid accumulation in the human brain after one night of sleep deprivation. Proc Natl Acad Sci U S A. 2018 Apr 24;115(17):4483-4488. doi: 10.1073/pnas.1721694115. Epub 2018 Apr 9. PubMed PMID: 29632177; PubMed Central PMCID: PMC5924922; 4: Chen DW, Wang J, Zhang LL, Wang YJ, Gao CY. Cerebrospinal Fluid Amyloid-β Levels are Increased in Patients with Insomnia. J Alzheimers Dis. 2018;61(2):645-651. doi: 10.3233/JAD-170032. PubMed PMID: 29278891; 5: Lucey BP, Hicks TJ, McLeland JS, Toedebusch CD, Boyd J, Elbert DL, Patterson BW, Baty J, Morris JC, Ovod V, Mawuenyega KG, Bateman RJ. Effect of sleep on overnight cerebrospinal fluid amyloid β kinetics. Ann Neurol. 2018 Jan;83(1):197-204. doi: 10.1002/ana.25117. PubMed PMID: 29220873; PubMed Central PMCID: PMC5876097; 6: Ju YS, Ooms SJ, Sutphen C, Macauley SL, Zangrilli MA, Jerome G, Fagan AM, Mignot E, Zempel JM, Claassen JAHR, Holtzman DM. Slow wave sleep disruption increases cerebrospinal fluid amyloid-β levels. Brain. 2017 Aug 1;140(8):2104-2111. doi: 10.1093/brain/awx148. PubMed PMID: 28899014; PubMed Central PMCID: PMC5790144; 7: Minakawa EN, Miyazaki K, Maruo K, Yagihara H, Fujita H, Wada K, Nagai Y. Chronic sleep fragmentation exacerbates amyloid β deposition in Alzheimer's disease model mice. Neurosci Lett. 2017 Jul 13;653:362-369. doi: 10.1016/j.neulet.2017.05.054. Epub 2017 May 26. PubMed PMID: 28554860; 8: Yulug B, Hanoglu L, Kilic E. Does sleep disturbance affect the amyloid clearance mechanisms in Alzheimer's disease? Psychiatry Clin Neurosci. 2017 Oct;71(10):673-677. doi: 10.1111/pcn.12539. Epub 2017 Jun 21. Review. PubMed PMID: 28523718; 9: Chen L, Huang J, Yang L, Zeng XA, Zhang Y, Wang X, Chen M, Li X, Zhang Y, Zhang M. Sleep deprivation accelerates the progression of alzheimer's disease by influencing Aβ-related metabolism. Neurosci Lett. 2017 May 22;650:146-152. doi: 10.1016/j.neulet.2017.04.047. Epub 2017 Apr 25. PubMed PMID: 28455102; 10: Kincheski GC, Valentim IS, Clarke JR, Cozachenco D, Castelo-Branco MTL, Ramos-Lobo AM, Rumjanek VMBD, Donato J Jr, De Felice FG, Ferreira ST. Chronic sleep restriction promotes brain inflammation and synapse loss, and potentiates memory impairment induced by amyloid-β oligomers in mice. Brain Behav Immun. 2017 Aug;64:140-151. doi: 10.1016/j.bbi.2017.04.007. Epub 2017 Apr 13. PubMed PMID: 28412140; 11: An H, Cho MH, Kim DH, Chung S, Yoon SY. Orexin Impairs the Phagocytosis and Degradation of Amyloid-β Fibrils by Microglial Cells. J Alzheimers Dis.2017;58(1):253-261. doi: 10.3233/JAD-170108. PubMed PMID: 28387679; 12: Farca Luna AJ, Perier M, Seugnet L. Amyloid Precursor Protein in Drosophila Glia Regulates Sleep and Genes Involved in Glutamate Recycling. J Neurosci. 2017 Apr 19;37(16):4289-4300. doi: 10.1523/JNEUROSCI.2826-16.2017. Epub 2017 Mar 17.PubMed PMID: 28314820; 13: Wei M, Zhao B, Huo K, Deng Y, Shang S, Liu J, Li Y, Ma L, Jiang Y, Dang L, Chen C, Wei S, Zhang J, Yang H, Gao F, Qu Q. Sleep Deprivation Induced Plasma Amyloid-β Transport Disturbance in Healthy Young Adults. J Alzheimers Dis. 2017;57(3):899-906. doi: 10.3233/JAD-161213. PubMed PMID: 28304302). In this sregard, amyloid beta has been widely linked to HPA axs dysfunction ( Neurobiol Stress. 2018 May 19;9:9-21. doi: 10.1016/j.ynstr.2018.05.003; Curr Pharm Des. 2014;20(15):2539-46; Front Aging Neurosci. 2018 Jul 24;10:204. doi: 10.3389/fnagi.2018.00204. Park HJ et al. EMBO J. (2015; Campbell SN et al. J Alzheimers Dis. (2015) and other similar) and it has been associated with depressive like phenotype either in animal model or human (Eur J Pharmacol. 2017 Dec 15;817:22-29. doi: 10.1016/j.ejphar.2017.08.031; J Alzheimers Dis. 2018;66(1):115-126. doi: 10.3233/JAD-180688.). Thus, I suggest to insert such part in order to increase the novelty of the work.

Author Response

We are glad about the favorable comments of this reviewer. It is an excellent suggestion to add a paragraph about the interaction of amyloid levels and sleep. Paragraph 6 is now followed by a new paragraph entitled “Amyloid-β and sleep”:

Aggregation and accumulation of amyloid-β (Aβ) contributes this development of Alzheimer’s disease (Mayeux & Stern, 2012). Several recent studies address the interaction of Aβ and sleep. Using positron emission tomography Shokri-Kojori et al. (2017) showed significant increases in Aβ burden in the right hippocampus and thalamus after a night of sleep deprivation in healthy controls. These increases were associated with worsening of mood after sleep deprivation. In rats, sleep deprivation impaired cognitive function and elevated Aβ levels (Chen et al., 2017). The effect of sleep on overnight cerebrospinal fluid (CSF) Aβ kinetics was tested in healthy volunteers using intracerebroventricular (icv) lumbar catheters for serial sampling of CSF while subjects were sleep deprived, received sleep promoting sodium oxybate or slept normally. To measure Aβ kinetics all participants were infused with 13 C 6-leucine. Sleep deprivation increased overnight Aβ38, Aβ40, and Aβ42 levels by 25-30 % via increased overnight Aβ production relative to sleeping subjects. The authors concluded that disrupted sleep increases Alzheimer’s disease risk by increased Aβ production (Lucey et al., 2018). In order to elucidate whether chronic sleep restriction potentiates the brain impact of Aβ oligomers (AβOs) studies in mice were performed. A single icv infusion of Aβ oligomers disturbed sleep pattern with rather chronically sleep restricted mice showed higher brain expression of pro-inflammatory mediators, reduced levels of pre- and post-synaptic marker proteins. Furthermore, this study exhibited increased susceptibility to the sub-toxic dose of AβOs on performance in a novel object recognition memory task. After sleep restriction,elevated brain tumor necrosis factor α (TNF- α) levels in response to AβOs. Neuronal impairment in sleep restricted AβOs infused mice was prevented by a TNF- α neutralizing monoclonal antibody. The authors discuss a dual relationship between sleep and Alzheimer’s disease with disruption of sleep wake patterns by AβOs and increased brain vulnerability to AβOs after chronic sleep restriction (Kincheski et al., 2017). In Alzheimer’s disease model mice, chronic sleep fragmentation was induced by a running-wheel-based device that resulted in increased Aβ deposition in the mouse brain. The severity of Aβ deposition showed a significant positive correlation with the extent of sleep fragmentation (Minakawa  et al., 2017).Specific disruption of SWA in healthy adults without sleep disorders correlated with an increase in Aβ (Ju et al., 2017).In patients with insomnia CSF Aβ levels were significantly higher than in healthy controls (Chen et al., 2018).

Interestingly, there is some overlap between the pathophysiology of depression, Alzheimer’s disease and sleep. Human neuroblastoma cells produced more Aβ after treatment with CRH (Park et al., 2015). Morgese et al. (2017) discuss that chronic stress may represent common biological bases linking Alzheimer’s dementia and depression. The interaction of sleep and Aβ in patients with depression is an open topic on the research agenda.

References

Mayeux, R.; Stern, Y. Epidemiology of Alzheimer disease. Cold Spring Harb. Perspect. Med. 2012, doi: https://doi.org/10.1101/cshperspect.a006239

Shokri-Kojori, E.; Wang, G.-J.; Wiers, C.E.; Demiral, S.B.; Guo, M.; Kim, S.W.; Lindgren, E.; Ramirez, V.; Zehra, A.; Freeman, C. β-Amyloid accumulation in the human brain after one night of sleep deprivation. Proceedings of the National Academy of Sciences 2018, 115, 4483-4488, doi:https://doi.org/10.1073/pnas.1721694115

Chen, L.; Huang, J.; Yang, L.; Zeng, X.-A.; Zhang, Y.; Wang, X.; Chen, M.; Li, X.; Zhang, Y.; Zhang, M. Sleep deprivation accelerates the progression of Alzheimer’s disease by influencing Aβ-related metabolism. Neuroscience letters 2017, 650, 146-152, doi:https://doi.org/10.1016/j.neulet.2017.04.047.

Lucey, B.P.; Hicks, T.J.; McLeland, J.S.; Toedebusch, C.D.; Boyd, J.; Elbert, D.L.; Patterson, B.W.; Baty, J.; Morris, J.C.; Ovod, V. Effect of sleep on overnight cerebrospinal fluid amyloid β kinetics. Ann. Neurol. 2018, 83, 197-204, doi:https://doi.org/10.1002/ana.25117.

Kincheski, G.C.; Valentim, I.S.; Clarke, J.R.; Cozachenco, D.; Castelo-Branco, M.T.; Ramos-Lobo, A.M.; Rumjanek, V.M.; Donato Jr, J.; De Felice, F.G.; Ferreira, S.T. Chronic sleep restriction promotes brain inflammation and synapse loss, and potentiates memory impairment induced by amyloid-β oligomers in mice. Brain. Behav. Immun. 2017, 64, 140-151, doi:https://doi.org/10.1016/j.bbi.2017.04.007.

Minakawa, E.N.; Miyazaki, K.; Maruo, K.; Yagihara, H.; Fujita, H.; Wada, K.; Nagai, Y. Chronic sleep fragmentation exacerbates amyloid β deposition in Alzheimer’s disease model mice. Neuroscience letters 2017, 653, 362-369, doi:https://doi.org/10.1016/j.neulet.2017.05.054.

Chen, D.-W.; Wang, J.; Zhang, L.-L.; Wang, Y.-J.; Gao, C.-Y. Cerebrospinal Fluid Amyloid-β Levels are Increased in Patients with Insomnia. J. Alzheimers Dis. 2018, 61, 645-651, doi:https://doi.org/10.3233/JAD-170032.

Park, H.J.; Ran, Y.; Jung, J.I.; Holmes, O.; Price, A.R.; Smithson, L.; Ceballos‐Diaz, C.; Han, C.; Wolfe, M.S.; Daaka, Y. The stress response neuropeptide CRF increases amyloid‐β production by regulating γ‐secretase activity. The EMBO journal 2015, 34, 1674-1686, doi:https://doi.org/10.15252/embj.201488795.

Ju, Y.-E.S.; Ooms, S.J.; Sutphen, C.; Macauley, S.L.; Zangrilli, M.A.; Jerome, G.; Fagan, A.M.; Mignot, E.; Zempel, J.M.; Claassen, J.A. Slow wave sleep disruption increases cerebrospinal fluid amyloid-β levels. Brain 2017, 140, 2104-2111, doi:https://doi.org/10.1093/brain/awx148.

Park, H.J.; Ran, Y.; Jung, J.I.; Holmes, O.; Price, A.R.; Smithson, L.; Ceballos‐Diaz, C.; Han, C.; Wolfe, M.S.; Daaka, Y. The stress response neuropeptide CRF increases amyloid‐β production by regulating γ‐secretase activity. The EMBO journal 2015, 34, 1674-1686, doi:https://doi.org/10.15252/embj.201488795.

Morgese, M.G.; Schiavone, S.; Trabace, L. Emerging role of amyloid beta in stress response: Implication for depression and diabetes. Eur. J. Pharmacol. 2017, 817, 22-29, doi:https://doi.org/10.1016/j.ejphar.2017.08.031.

Reviewer 3 Report

The authors reviewed depression and sleep. This paper is interesting and useful. However, the authors have the tendency to mention only the clinical findings of depression and sleep except for the HPA axis. I would like the authors to discuss the following mechanisms in detail.

1)      Risk Gene

What is the function of P2RX protein? How the amino acid and structural change is supposed to change the clinical symptom of both depression and sleep pattern?

2)      HPA axis

How do antidepressants e.g. SSRI induce HPA axis? What mechanism is supposed that cortisol induces depressive symptoms and the increase of REM?

Author Response

We are glad for this reviewer’s favorable comment.

We address this reviewer’s points as follows.

1.Risk Gene

o   What is the function of P2RX protein? How the amino acid and structural change is supposed to change the clinical symptom of both depression and sleep pattern?

We added to § 4 of the manuscript, line 107:

PXX7R is found in immune, endothelial, and epithelial cells and regulates various aspects of immune function, as expression and secretion of cytokines (Wiley et al., 2011).

Addition to § 4, line 108:

(base change 1405A>G) leading to substitution of glutamine (Gln, Q) biarginine (Arg, R), at codon 460 (Gln 460 Arg, Q 460 R) has been associated with mood disorders.

At the end of line 119 we added some lines about related animal studies.

“Mice that harbor P2X7R-Gln 460 AG and the wild-type P2X7R showed, compared to homozygous P2X7R wildtype and P2X7RhQ460R mice an increase of entries to REM sleep during the light period, suggesting a stronger drive towards REM sleep and more fragmented sleep cycles. Furthermore, SWA was lower and the amount of deep non-REM sleep was only small in heterozygous mice. Taken together heterozyeous mice show altered sleep architecture and reduced sleep quality compared to homozygous mice (Metzger et al., 2018).”

We feel that these additions enrich this paragraph. It is not possible to say how the amino acid and structural chain result in changes of the clinical symptoms of depression. An episode of depression in a patient is always triggered by an interaction of various genes and the environment. Similarly, it is not possible to describe a mechanism how the structural change in the SNP influences the sleep pattern. The interesting observations are that totally healthy carriers of one risk gene show subtle sleep-EEG changes and that particularly heterozygosity for the wildtype and its mood disorder associated variant represent a genetic risk factor.

2.“How do antidepressants e.g. SSRI induce HPA axis?”  

We have added the citation of an excellent review which deals with many aspects of the role of the HPA system in depression to line 166 (Holsboer & Ising 2010).

 It is thought that glucocorticoid receptors are downregulated in depression following stress exposure, whereas antidepressants upregulate mineralocorticoid and glucocorticoid receptors. This topic goes beyond the scope of this manuscript. Interested readers should refer to the cited review.

“What mechanism is supposed that cortisol induces depressive symptoms and the increase of REM?”

Probably, cortisol does not induce depressive symptoms. Elevated cortisol is the endpoint of overactivity of the HPA system. The key hormone of the HPA system is CRH which impairs sleep and promotes REM sleep. Our study on the sleep-EEG effects of high dosages of glucocorticoids (line 208) suggests a synergism of CRH and cortisol in the promotion of REM sleep. To clarify this issue, we added to line 221 “A synergism of CRH and cortisol may contribute to REM sleep disinhibition.”

References

Holsboer, F.; Ising, M. Stress Hormone Regulation: Biological Role and Translation into Therapy. Annu. Rev. Psychol. 2010, 61, 81-109, doi:10.1146/annurev.psych.093008.100321.

Wiley, J.; Sluyter, R.; Gu, B.; Stokes, L.; Fuller, S. The human P2X7 receptor and its role in innate immunity. Tissue Antigens 2011, 78, 321-332, doi:https://doi.org/10.1111/j.1399-0039.2011.01780.x.

Round 2

Reviewer 3 Report

none